# Measurement of Fracture Toughness of Pure Tungsten Using a Small-Sized Compact Tension Specimen

**DOI:** 10.3390/ma13010244

**Published:** 2020-01-06

**Authors:** Byeong Seo Kong, Ji Ho Shin, Changheui Jang, Hyoung Chan Kim

**Affiliations:** 1Department of Nuclear and Quantum Engineering, Korea Advanced Institute of Science and Technology, 291 Daehak-ro, Yuseong-gu, Daejeon 34141, Korea; assaultpc@kaist.ac.kr (B.S.K.); shinjiho@kaist.ac.kr (J.H.S.); 2National Fusion Research Institute, 169-148 Gwahak-ro, Yuseong-gu, Daejeon 34133, Korea; chankim@nfri.re.kr

**Keywords:** tungsten, fatigue pre-crack, fracture toughness test

## Abstract

The evaluation of fracture toughness of pure tungsten is essential for the structural integrity analysis in a fusion reactor. Therefore, the accurate quantification of fracture toughness of tungsten alloys is needed. However, due to the inherent brittleness of tungsten, it is difficult to introduce a sharp fatigue pre-crack needed for the fracture toughness test. In this study, a novel fatigue pre-cracking method was developed and applied to the small-sized disc-type compact tension (DCT) specimens of double-forged pure tungsten. To overcome the brittleness and poor oxidation resistance, a low-frequency tensile fatigue pre-cracking was performed at 600 °C in Ar environment, which resulted in the introduction of a sharp pre-crack to DCT specimens. Then, fracture toughness tests were conducted at room temperature (RT), 400 °C, and 700 °C in air and Ar gas environments using as-machined and pre-cracked DCT specimens. At RT and 400 °C, the test environment and crack tip radius did not affect the fracture toughness measurement. However, at 700 °C, the Ar gas environment and the presence of a sharp fatigue pre-crack resulted in a decrease in the measured fracture toughness. Thus, it was suggested that, for the conservative fracture toughness measurement of pure tungsten, fatigue pre-cracking and fracture toughness test should be performed in an inert environment, especially for high-temperature tests.

## 1. Introduction

Pure tungsten (W) is considered for plasma-facing materials of the fusion reactors including diverter components [1,2,3,4]. For fusion applications, tungsten has various advantages such as high melting point, high thermal conductivity, low tritium retention, and low material erosion for application as diverter materials [1]. Meanwhile, it has relatively high ductile to brittle transition temperature (DBTT), poor oxidation resistance at high temperature, low recrystallization temperature, and inherent brittleness [1,5,6,7]. The inherent low fracture toughness of tungsten and resulting poor fracture resistance could be limiting factors in structural materials application [1,3]. Therefore, evaluation of fracture toughness of W is essential to structural integrity analysis of W diverters. Nonetheless, published fracture toughness data and related studies are rare in current literature [5,8].

For fracture toughness tests of metallic materials, a sharp crack should be introduced to the specimen [9,10,11] to simulate the cracks in the components. Generally, in the case of typical metallic materials, a sharp crack could be easily produced by applying tensile cyclic loading, or by fatigue pre-cracking, to the machine-notched specimen. Meanwhile, for extremely hard metals like tungsten, this conventional method would not work due to the high stresses required to initiate crack and the very small stresses needed for driving a sharp crack [12]. If the notch root radius of inserted pre-crack is not sharp enough, the measured fracture toughness could be non-conservative and may not be used for integrity analysis. Thus, a number of alternative methods have been proposed because control of fatigue crack propagation is extremely difficult in brittle materials [12,13,14]. For metallic/ceramic functionally graded materials, reverse four-point bending loadings were used to introduce a sharp pre-crack [12], but such technique requires relatively wide specimens. Similarly, compressive fatigue cycles were applied to bar specimens of WC-Co alloy to produce a sharp pre-crack, which was enclosed by semicircular cracks due to local deformation [13]. Also, ultra-short pulse laser was successful to introduce a pre-crack with enough sharpness, but there was a concern for the high cost of such a method [14].

Thus, in this study, a novel method of introducing a sharp pre-crack to small-sized tungsten specimens was developed. In order to introduce a sharp pre-crack to brittle W specimens with good reproducibility, tensile fatigue pre-cracking was performed at 600 °C in Ar gas environment. Then, fracture toughness of pure tungsten was measured at room temperature (RT), 400 °C, and, 700 °C using as-machined and pre-cracked specimens to evaluate the effect of the notch root radius. Also, fracture toughness tests were performed in air and Ar environment to evaluate the effect of environment on fracture toughness measurement. Finally, the proper pre-cracking method and test environment will be suggested for fracture toughness measurement of pure tungsten.

## 2. Materials and Methods

### 2.1. Materials and Specimen

The pure tungsten investigated in this study is a double-forged pure tungsten (DFW) rod manufactured by Plansee AG. The chemical compositions are listed in Table 1. The typical microstructure of DFW is shown in Figure 1, which was taken by electron backscatter diffraction (EBSD, EDAX Inc., Mahwah, NJ, USA) from a field emission scanning electron microscope (FE-SEM; JSM-7100F, JEOL, Tokyo, Japan) equipped with orientation imaging microscopy (OIM) software (EDAX OIM v7). As shown in Figure 1a, a significant amount of sub-grains with small misorientation within certain crystallographic orientations are found inside the larger grains, as indicated by color contrast inside of grains. The measured grain boundary character distribution is presented in Figure 1b. Since sub-grains are defined by low angle grain boundary (LAGB), around 80% of the grain boundaries are those of sub-grains in this condition. The measured average grain size including both high angle grain boundary (HAGB) and LAGB is approximately 3.1 μm. Tensile properties of DFW were measured at several temperatures and all tests were conducted at a crosshead speed of 0.2 mm/min corresponding to a strain rate of 2 × 10^−4^/s. The elongation was measured by a change of displacement without the use of an extensometer due to the small size of the specimen. The results of tensile tests are shown in Table 2.

The fracture toughness test specimen used in this study is a small-sized disc compact (DCT) specimen shown in Figure 2. Specimens were fabricated by electro-discharge machining (EDM) in such a way that the crack plane was parallel to the longitudinal direction and the crack direction was in the circumferential direction of the DFW rod. The geometry of the DCT specimen complied with general proportions of standard configuration of ASTM-E399 and ASTM-E1820. The dimensions of DCT specimens are 8.1 mm of outer diameter, 6 mm of width (W), 1.5 mm of thickness (B), 2.7 mm of machined notch (a_0_), and 3.3 mm of remaining ligament (b_0_). The thickness of the specimen was chosen to satisfy the thickness required for the linear elastic [9] or elastic-plastic fracture toughness tests [10]. The notch root radius of the machined notch was approximately 70 μm.

### 2.2. Fatigue Pre-Cracking

The DCT specimens with machined notch were fatigue pre-cracked under tensile cyclic loading. To ensure stable fatigue crack growth under tensile cyclic loading, pre-cracking was performed at 600 °C, where a certain amount of ductility would be provided. To prevent the excessive plastic zone formation in front of fatigue pre-crack, the maximum fatigue load was limited by Equation (1) as follows:(1)$$P_{m}=\frac{0.4{\mathrm{Bb}}_{0}^{2}\sigma_{Y}}{2W+a_{0}}$$
where P_m_ is maximum allowed fatigue load, σ_Y_ is flow stress, a_0_ is initial crack length or length of machined notch. Using σ_Y_ of 468 MPa (from Table 2) and dimensions, P_m_ was calculated as 217 N. In the pre-cracking, the maximum tensile load of 212 N was used with R-ratio (P_max_/P_min_) = 0.1. P_min_ corresponds to a minimum tensile load in cyclic load during the fatigue pre-cracking. The first trial of fatigue pre-cracking was carried out at 600 °C in air for approximately 5000 cycles to achieve ~200 μm of fatigue pre-crack, resulting in a/W ~ 0.5. However, it was found that an extensive oxide layer was found in the pre-crack and the resulting crack tip was not sharp enough (to be described later).

To prevent excessive oxidation during fatigue pre-cracking, some specimens were fatigue pre-cracked in Ar environment within the specially designed quartz tube as shown in Figure 3. To maintain the inert environment, Ar was continuously injected into the quartz tube and the specimen was located close to Ar gas inlet and far away from the sealing where the leak could happen. It was confirmed that oxidation could be minimized at the Ar flow rate of 700 cc/min for the 2 h duration of pre-cracking. To confirm the effectiveness of the pre-cracking method, the fatigue pre-crack area was observed using a scanning electron microscope (SEM, Hitachi, Tokyo, Japan) and the crack tip root radius was also measured for the specimen pre-cracked in air and Ar gas environment.

### 2.3. Fracture Toughness Tests

The fracture toughness tests were performed using a universal mechanical testing machine (Instron 4204, load capacity ~ 50 kN) at RT, 400 °C, and 700 °C with a cross-head speed of 0.3 mm/min. As summarized in Table 2, the DFW failed with small elongation and showed rather brittle failure at 400 °C, whereas elongation was large at 600 °C or higher. Therefore, it was expected that brittle fracture would be observed at RT and 400 °C while ductile fracture at 700 °C. Therefore, at RT and 400 °C, fracture toughness value was calculated following ASTM-E399 based on linear elastic fracture mechanics (LEFM). Meanwhile, at 700 °C, J-R (J-integral verse resistance) tests were performed using the normalization method described in ASTM-E1820 by means of elastic–plastic fracture mechanics (EPFM). Also, fracture toughness tests were performed in air and Ar environments to check the effect of test environment. As a results, a series of test conditions with different pre-cracking and fracture toughness test environments were classified, namely as as-machined and tested in air (MA), pre-cracked in air and tested in air (AA), pre-cracked in Ar gas environment and tested in air (GA), and pre-cracked in Ar gas environment and tested in Ar gas environment (GG). After the tests, tentative fracture toughness values, such as K_Q_ and J_Q_, were calculated and the thickness requirements were checked. Finally, the fracture surface of the GG specimen tested at 700 °C was observed under SEM to evaluate the cracking mode.

## 3. Results and Discussion

### 3.1. Fatigue Pre-Crack Produced at 600 °C

The morphologies of fatigue pre-cracks produced in the air and the Ar gas environment are presented in Figure 4. For both cases, fatigue pre-cracks are quite straight and ~200 μm in depth. It is clear that the inside of the pre-crack produced at 600 °C in air is filled with oxides and the notch root radius is estimated ~1 μm (Figure 4a). On the other hand, for the specimen pre-cracked at 600 °C in the Ar gas environment, fatigue pre-crack is quite narrow without oxides and its tip is very sharp. In this case, the notch root radius is estimated ~0.2 μm (Figure 4b), much smaller than that of air pre-cracked specimen. Thus, it was shown that sharp fatigue pre-crack could be successfully produced for DCT specimens of DFW by proper control of temperature and environment.

### 3.2. Fracture Toughness Test Results

The results of fracture toughness tests at RT and 400 °C are represented in Figure 5 and summarized in Table 3. The measured fracture toughness is around 8 MPam^1/2^ at RT and 30 MPam^1/2^ at 400 °C, respectively. These fracture toughness values are similar to previously published data [1,15,16]. It should be noted that there is no clear effect of fatigue pre-cracking, pre-cracking environment, or test environment for fracture toughness at 400 °C as shown in Figure 5a. From the load-displacement curves in Figure 5b, it is clear that DFW would show linear elastic fracture behavior at 400 °C for all pre-cracking conditions and test environments. Thus, in the brittle fracture regime of DFW, the fracture toughness test using specimens with as-machined notch in air may provide reasonably accurate fracture toughness values.

On the other hand, at 700 °C, there are significant differences in measured fracture toughness values (J_Q_) depending on fatigue pre-cracking, pre-cracking environment, and test environment as summarized in Table 3. Also, as shown in Figure 6, the J-R curves and load-displacement curves are quite different among specimen classes. From Figure 6a, measured fracture resistance is larger in the order or MA ~ AA > GA > GG. For fracture toughness value estimation, the blunting line was constructed by J = Mσ_Y_Δa, in which the expression of a value of 2 or 4 for M was used to determine J_Q_ value [10,17]. Meanwhile, it was reported that the slope of the blunting line (Mσ_Y_) for determination of J_Q_ may not affect when the ratio of yield strength (σ_YS_) to tensile strength (σ_UTS_) is close to 1 [18]. As summarized in Table 2, the calculation result of σ_YS_/σ_UTS_ at 700 °C is 0.95, which is almost 1. So, either of M values could be used to find J_Q_ in this study. On the other hand, due to the steep slope of the blunting line, it was difficult to define the intersection points for MA and AA conditions. Therefore, in order to compare all of the data sets (MA, AA, GA, and GG), we chose an M value of 4 to estimate fracture toughness values.

As shown in Table 3, there is significant difference in measured fracture toughness between MA (J_Q_ ~ 280 kJ/m^2^) and GG (J_Q_ ~ 61 kJ/m^2^) conditions, suggesting the importance of careful control of fatigue pre-cracking and test environment for DFW which has poor oxidation resistance at high temperature. The load-displacement curves (Figure 6a) show similar behavior, such that the MA specimens show much extended load-displacement curves compared to the GG specimens which show very rapid load drop after maximum load.

When DCT specimens were tested in air, the fracture toughness of air pre-cracked specimen (AA) is not much different from that of the as-machined specimen (MA), suggesting that oxidation during test is dominant factor rather than the difference in notch root radius. Meanwhile, in the case of DCT specimens pre-cracked in the Ar gas environment, a decrease in fracture toughness was apparently observed even in the air environment (GA) compared to MA and AA conditions. These results indicate that with a very sharp pre-crack, the effect of oxidation during the test could be somewhat alleviated, though not completely. Yet, the most conservative fracture toughness values of DFW can be measured when DCT specimens with sharp pre-crack are tested in an inert environment like the Ar gas environment (GG). In an oxidizing environment, crack tip blunting could be associated with oxidation at the crack tip at high temperatures [15,19]. Also, enhanced oxidation assisted by high stress at the crack tip could affect the onset of crack growth during J-R tests, resulting in longer stable crack growth as observed for specimens tested in air. Such behavior was more extensive when blunted crack was already present before the test as MA and AA conditions (Figure 6). Therefore, the testing environment at high temperature is important not only for fatigue pre-cracking but also for J-R tests for fracture toughness measurement of DFW.

For calculated tentative fracture toughness (J_Q_), the size requirement for valid fracture toughness values was checked. According to the criteria of ASTM-E1820, thickness (B) and initial ligament (b_0_) should be greater than 10 J_Q_/σ_Y_. It was calculated that, for a given geometry and tensile properties, the measured fracture toughness (J_Q_) should be less than 67.3 kJ/m^2^ to meet the size requirement. As presented in Table 3, the only condition that satisfies size requirement is when the DCT specimen pre-cracked in the Ar environment was tested in the Ar gas environment (GG).

### 3.3. Fracture Surface Observation for Specimen Tested at 700 °C

Figure 7 shows the SEM micrographs of the fracture surface of J-R tested DCT specimen (GG specimen). As shown in Figure 7a, approximately 200 μm of fatigue pre-crack is clearly observed without oxides at the surface. At higher magnification (Figure 7b), the transition from fatigue pre-crack to stable crack growth was apparently distinguished. In the pre-crack region (Figure 7c), some cleavage facets, intergranular cleavage, and faceted striation are observed, which indicate minimum plastic deformation during tensile fatigue pre-cracking. Meanwhile, ductile dimple fracture morphology is evident in stable crack growth region during J-R test with static loading [8].

### 3.4. Implication to Fracture Toughness Test of Tungsten Alloys

The novel pre-cracking method developed in this work could be used to introduce a sharp fatigue pre-crack to various tungsten alloys and potentially to other very brittle and poor oxidation resistant materials. The clear advantage of the current method is utilizing tensile fatigue loadings instead of much troublesome compressive fatigue loadings, can thus be easily applicable to compact tension-type specimens of small size. Nonetheless, for such applications, detailed pre-cracking temperature and environment should be carefully chosen to avoid excessive cracking or crack-tip blunting by oxidation.

It was shown that at temperatures in which DFW shows brittle fracture behavior, the presence of a sharp pre-crack or the test environment did not affect the measured fracture toughness as shown in Table 3. However, at higher temperature in which DFW shows ductile behavior, the sharp pre-crack and the test environment greatly affect the fracture toughness measurement, such that specimens with a sharp fatigue pre-crack tested in an inert environment (GG condition) showed the lowest, or the most conservative fracture toughness value (Table 3). Thus, it was suggested that, for the conservative fracture toughness measurement of pure tungsten, fatigue pre-cracking and fracture toughness test should be performed in an inert environment, especially for high-temperature tests. Also, the developed pre-cracking method could be used to the more extensive evaluation of the fracture behavior of tungsten alloys in wide temperature ranges and various environments.

## 4. Conclusions

A novel pre-cracking method for pure tungsten, a brittle and poor oxidation-resistant material, was developed. To overcome the brittleness and oxidation, tensile fatigue pre-cracking was performed at 600 °C in Ar gas environment, which successfully produced a sharp fatigue pre-crack of ~200 μm in depth. Then, fracture toughness tests were conducted at RT, 400 °C, and 700 °C in air and Ar gas environments using as-machined and pre-cracked DCT specimens. At RT and 400 °C, the test environment and crack tip radius did not affect the fracture toughness measurement. However, at 700 °C, the Ar gas environment and the presence of a sharp fatigue pre-crack resulted in conservative fracture toughness values. Thus, it was suggested that, for the conservative fracture toughness measurement of pure tungsten, fatigue pre-cracking and fracture toughness test should be performed in an inert environment, especially for high-temperature tests.

## Figures and Tables

**Figure 1 materials-13-00244-f001:**
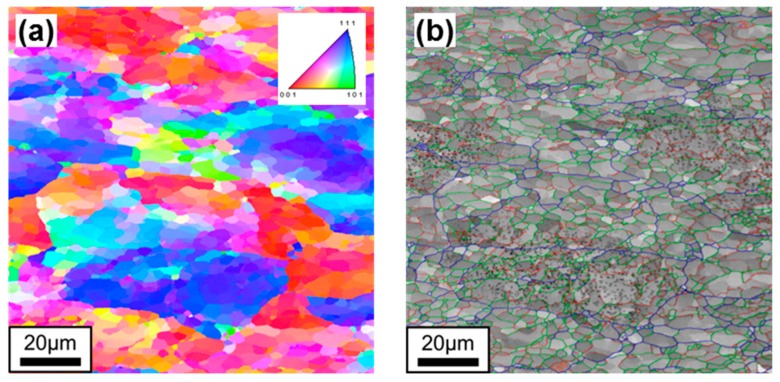
The results of electron backscatter diffraction (EBSD) analysis for DFW: (**a**) inverse pole figure (IPF) map and (**b**) grain boundary character, where blue, green, and red lines indicate high misorientation angle boundaries (15° ≤ θ < 180°) and low misorientation angle boundaries (5° ≤ θ < 15°, 2° ≤ θ < 5°), respectively. (Gray spots in (**b**) are residue of colloidal silica used in polishing.).

**Figure 2 materials-13-00244-f002:**
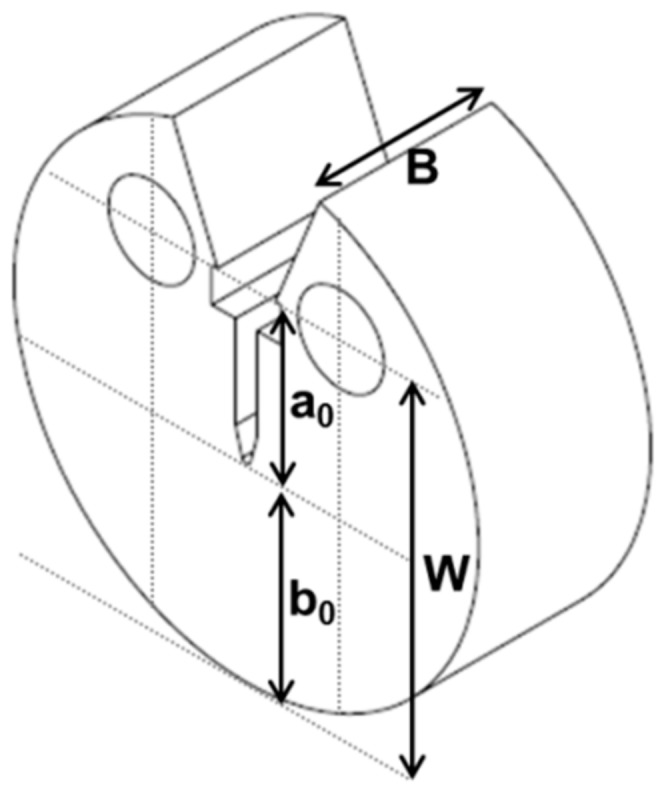
The schematics of the disc compact (DCT) specimen for fracture toughness test.

**Figure 3 materials-13-00244-f003:**
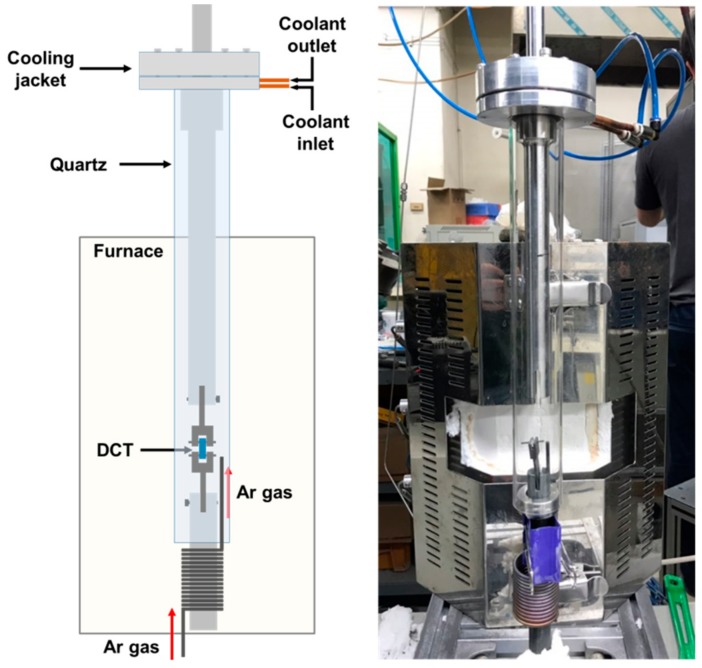
The schematics of fatigue pre-cracking set-up in high-temperature Ar environment.

**Figure 4 materials-13-00244-f004:**
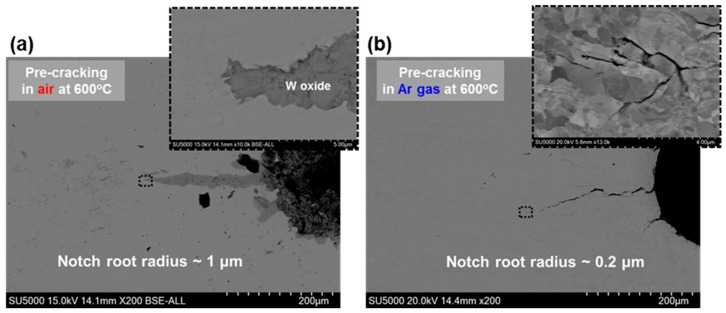
The morphologies of fatigue pre-cracks of the DCT specimen; (**a**) pre-cracked at 600 °C in air, (**b**) pre-cracked at 600 °C in Ar gas environment.

**Figure 5 materials-13-00244-f005:**
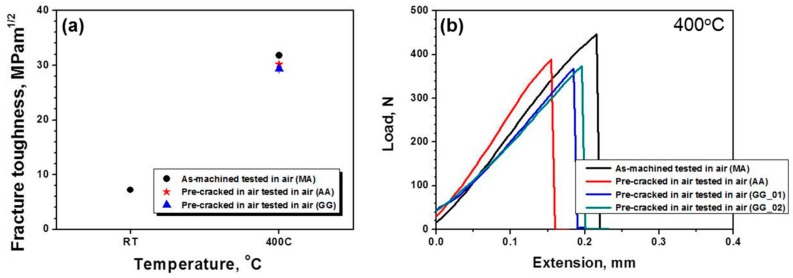
(**a**) The results of fracture toughness test at room temperature (RT) and 400 °C and (**b**) the load-displacement curves for specimens tested at 400 °C.

**Figure 6 materials-13-00244-f006:**
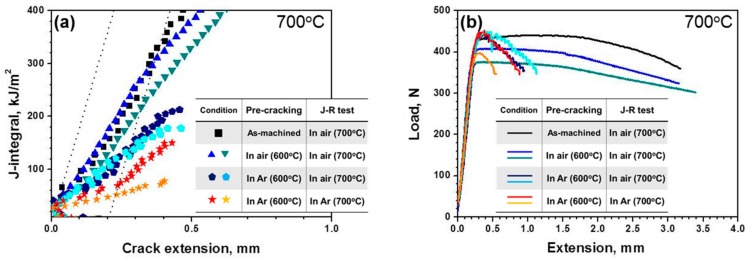
The results of fracture toughness test at 700 °C; (**a**) the load-extension curves, (**b**) the J-integral-crack extension curves.

**Figure 7 materials-13-00244-f007:**
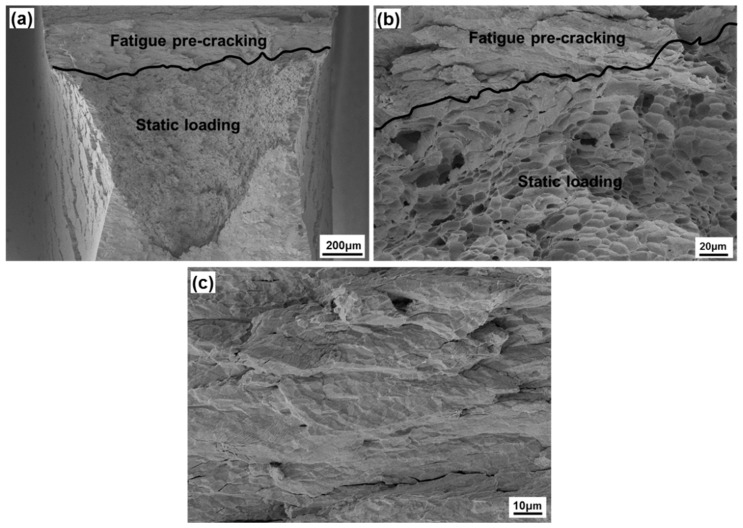
Fracture surface of J-R tested DCT specimen (pre-cracked at 600 °C and tested at 700 °C in Ar gas environment): (**a**) low magnification, (**b**) details of transition area from fatigue pre-crack to stable crack growth, and (**c**) details of fatigue pre-cracked region.

**Table 1 materials-13-00244-t001:** Chemical composition of as-received double-forged pure tungsten (DFW) rod (10^−3^ wt.%).

Chemical Composition	W	Cu	Mo	Cr	C
As-received DFW	Balance	17.0	3.9	2.5	1.1

**Table 2 materials-13-00244-t002:** Tensile properties of DFW rod.

Test Temperature	Yield Strength	Tensile Strength	Elongation
	(MPa)	(MPa)	(%)
400 °C	487.9 ± 10.7	510.7 ± 2.9	6.9 ± 2.4
600 °C	454.0 ± 16.9	482.0 ± 2.3	22.6 ± 0.2
700 °C	437.0 ± 18.4	460.0 ± 8.5	27.2 ± 7.3

**Table 3 materials-13-00244-t003:** Fracture toughness of DFW under different pre-cracking and testing conditions.

Pre-Cracking Condition	Testing Condition			
	Fracture Toughness (K_Q_)		Fracture Toughness (J_Q_)	
	$\mathrm{at}\;400\;{}^{\circ}C\;[\mathrm{MPa}\sqrt{m}]$		at 700 °C [kJ/m^2^]	
	In Air	In Ar Gas	In Air	In Ar Gas
As-machined	31.8	-	279.6	-
Pre-cracking in air	30.2	-	252.6 ± 55.6	-
Pre-cracking in Ar gas	-	29.3 ± 0.3	126.2 ± 2.5	61.1 ± 13.1

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
