# Peer review of "Measurement of Fracture Toughness of Pure Tungsten Using a Small-Sized Compact Tension Specimen"

_materials, 2020, doi:10.3390/ma13010244_

Round 1
Reviewer 1 Report
This paper presented a new pre-cracking method for measuring the fracture toughness of pure tungsten using small-sized compact tension specimens. In general, the manuscript is well written. The detailed comments on the submission are as follows:
The state of art was not clearly introduced and please elaborate. The abstract and introduction need to be improved to emphasize the novelty of the manuscript. The authors discussed the measurements at room temperature, 400ºC, and 700ºC. Please explain why these three temperatures were chosen. At the end of the paper, a reader wants to know how to benefit from the work you accomplished in your paper. Also, the limitations must be discussed by the authors and to the related need for further work. Page 3, line 99, what is the difference between “flow stress” and “Yield strength” in table 2. How the authors choose 468 MPa for the flow stress? Please explain. Please explain “Bal.” in table 1. Do not use bold format for the Figures and Tables in the main content. please check the manuscript thoroughly. Figure 4, there are two zoom-in pictures in the top right corner of figure 4 (a) and (b), but they do not seem to match the marked areas of the original figures. Please explain. Figure 5, the legend of figure 5 (b) is not correct, it should be MA, AA, GA, GG not MA, AA, GG, GG. Table 3, why the result of “Pre-cracking in Ar gas tested in air ” is not available at 400ºC. Figure 6, the picture quality is not good. The label of the y-axis of figure 6 (b) was blocked. Page 5, line 146, “data [1,15,16]”, please use black color instead of red for “15,16”. Page 6, line160, “in which expression a value of 4 for M was used for in this study”, please elaborate on why the authors choose 4.Author Response
We appreciate the valuable comments on the manuscript from the experts on measurement of fracture toughness and tungsten materials. We revised the manuscript reflecting the comments as best as possible. Here are the answers to the comments and the changes are marked in red in the revised manuscript.
Answers to REVIEWER 1’s Comments:
Comment 1.1. The state of art was not clearly introduced and please elaborate. The abstract and introduction need to be improved to emphasize the novelty of the manuscript.
Ans.: As reviewer suggested, the following sentences were added to Abstract to emphasize the novelty of the study.
“Therefore, the accurate quantification of fracture toughness of tungsten alloys is needed.”
“To overcome the brittleness and poor oxidation resistance, a low frequency tensile fatigue pre-cracking was performed at 600oC in Ar environment, which resulted in introduction of a sharp pre-crack to DCT specimens.
Also, the following sentences were added to the last paragraph of Introduction to emphasize the novelty of the study.
“Thus, in this study, a novel method of introducing a sharp pre-crack to small-sized tungsten specimens was developed. In order to introduce a sharp pre-crack to brittle W specimens with good reproducibility, tensile fatigue pre-cracking was performed at 600oC in Ar gas environment.”
Also, the following sentences were added to the end of the 2nd paragraph of Introduction to describe more of the state-of-the-art of fatigue pre-cracking for fracture test of tungsten.
“For metallic/ceramic functionally graded materials, reverse four-point bending loadings were used to introduce a sharp pre-crack [12], but such technique requires relatively wide specimens. Similarly, compressive fatigue cycles were applied to bar specimens of WC-Co alloy to produce a sharp pre-crack, which was enclosed by semicircular cracks due to local deformation [13]. Also, ultra-short pulse laser was successful to introduce a pre-crack with enough sharpness, but there was concern for the high cost of such method [14].”
Comment 1.2. The authors discussed the measurements at room temperature, 400ºC, and 700ºC. Please explain why these three temperatures were chosen.
Ans.: We intended to test the double forged tungsten (DFW) in brittle and ductile fracture regime to see the effect of a sharp fatigue pre-crack and test environment. Figure R1 (not shown in the manuscript) shows the results of tensile test for DFW depending on test temperature. The DFW seems to show rather brittle behavior up to 400oC but ductile behavior at higher temperature (600oC ~ 800oC). Thus, it was estimated that the ductile-to-brittle-transition temperature (DBTT) would be between 400oC and 600oC. So, we chose RT and 400ºC as brittle fracture, and 700ºC as ductile fracture property measurement. To clarify the reason of choosing the test temperatures, the following statements are added in Sec. 2.3.
“As summarized in Table 2, the DFW failed with small elongation and showed rather brittle failure at 400oC, whereas elongation was large at 600oC or higher. Therefore, it was expected that brittle fracture would be observed at RT and 400oC while ductile fracture at 700oC.”
Figure R1. The stress-strain curves of DFW depending on test temperature (not shown in manuscript)
Comment 1.3. At the end of the paper, a reader wants to know how to benefit from the work you accomplished in your paper. Also, the limitations must be discussed by the authors and to the related need for further work.
Ans.: As reviewer suggested, we added a new section before the conclusion to describe the benefit and limitation of current method.
“3.4. Implication to fracture toughness test of tungsten alloys
The novel pre-cracking method developed in this work could be used to introduce a sharp fatigue pre-crack to various tungsten alloys and potentially to other very brittle and poor oxidation resistant materials. The clear advantage of current method is utilizing tensile fatigue loadings instead of much troublesome compressive fatigue loadings, thus can be easily applicable to compact tension-type specimens of small size. Nonetheless, for such applications, detailed pre-cracking temperature and environment should be carefully chosen to avoid excessive cracking or crack-tip blunting by oxidation.
It was shown that at temperatures in which DFW shows brittle fracture behavior, the presence of a sharp pre-crack or the test environment did not affect the measured fracture toughness as shown in Table 3. However, at higher temperature in which DFW shows ductile behavior, the sharp pre-crack and the test environment greatly affect the fracture toughness measurement, such that specimens with a sharp fatigue pre-crack tested in inert environment (GG condition) showed the lowest, or the most conservative fracture toughness value (Table 3). Thus, it was suggested that, for the conservative fracture toughness measurement of pure tungsten, fatigue pre-cracking and fracture toughness test should be performed in inert environment, especially for high temperature tests. Also, the developed pre-cracking method could be used to the more extensive evaluation of the fracture behavior of tungsten alloys in wide temperature ranges and various environments.”
Comment 1.4. Page 3, line 99, what is the difference between “flow stress” and “Yield strength” in table 2. How the authors choose 468 MPa for the flow stress? Please explain.
Ans.: According to pre-cracking procedure of ASTM standards, the maximum fatigue load was limited by Eq. (1). In Eq. (1), flow stress is defined as a mean value between yield strength and tensile strength in ASTM-E1820. Thus, in case of DFW at 600oC, the flow stress is calculated as average of 454.4 MPa and 482.0 MPa. Therefore, flow stress is around 468 MPa.
Comment 1.5. Please explain “Bal.” in table 1. Do not use bold format for the Figures and Tables in the main content. please check the manuscript thoroughly.
Ans.: As the review pointed out, bold characters in main text has been changed to normal format. In table 1, the meaning of “Bal” is abbreviation of balance. The weight percent (wt. %) of W indicated by “Bal.” is considered as the remaining amount excluding the sum of Cu, Mo, Cr, and C. As the reviews commented, the Table 1 in Section 2.1 was modified as follows.
Table 1. Chemical composition of as-received double-forged pure tungsten (DFW) rod (10-3 wt.%)
Chemical composition |
W |
Cu |
Mo |
Cr |
C |
As-received DFW |
Balance |
17.0 |
3.9 |
2.5 |
1.1 |
Comment 1.6. Figure 4, there are two zoom-in pictures in the top right corner of figure 4 (a) and (b), but they do not seem to match the marked areas of the original figures. Please explain.
Ans.: Because we used pictures of large difference in magnification, it was not easy to match zoom-in pictures with marked areas, especially for Figure 4(b). As indicated, the enlarged area is much smaller than the area shown as marked area. We intended to show the locations where the zoom-in pictures are taken. In the revised manuscript, we reduced the size of marked area as much as possible. For pre-crack introduced in Ar gas atmosphere, the pre-crack tip is too tiny and narrow to be clearly observed in low magnification in Figure 4(b). To indicate the location of zoom-in picture was taken, picture of intermediate magnification was added in Figure R2 (not shown in the manuscript).
Figure R2. (Figure 4 (b) in the manuscript) The morphologies of fatigue pre-cracks of DCT specimen at 600oC in Ar gas environment.
Comment 1.7. Figure 5, the legend of figure 5 (b) is not correct, it should be MA, AA, GA, GG not MA, AA, GG, GG. Table 3, why the result of “Pre-cracking in Ar gas tested in air” is not available at 400ºC.
Ans.: In Figure 5 (b), there are two test results of GG. In the revised manuscript we indicated them as GG_01 and GG_02 in Figure R3 (Figure 5 (b) in the revised manuscript).
In Table 3, the result of “Pre-cracking in Ar gas tested in air (GA)” at 400ºC is not provided as we did not test such specimen condition. Part of the reason was that we did not have enough specimens to test all conditions. Another, more practical reason was that as we did not expect the test result would be significantly different from less conservative conditions (MA and AA) and most conservative condition (GG). At brittle fracture regime of 400 ºC, measure fracture toughness values were all very similar as shown in Table 3.
Figure R3. (Figure 5 in the revised manuscript) (a) The results of fracture toughness test at RT
and 400 oC and (b) the load-displacement curves for specimens tested at 400 oC.
Comment 1.8. Figure 6, the picture quality is not good. The label of the y-axis of figure 6 (b) was blocked.
Ans.: As the reviewer pointed out, we rearranged the Figure 6 to show the label of the y-axis of Figure 6 (b). In addition, we converted image as TIF format to get better image quality.
Figure R4. (Figure 6 in the revised manuscript) The results of fracture toughness test at 700oC; (a) the load-extension curves, (b) The J-integral-crack extension curves.
Comment 1.9. Page 5, line 146, “data [1,15,16]”, please use black color instead of red for “15,16”.
Ans.: As the reviewer pointed out, the color of the numbers indicating references was corrected to black.
Comment 1.10. Page 6, line160, “in which expression a value of 4 for M was used for in this study”, please elaborate on why the authors choose 4.
Ans.: The blunting line is defined as J=MσYΔa, where M is the constraint factor, σY is the flow stress, Δa is crack extension. In general, a value of 2 for M represented in ASTM-E1820 is widely used to determine JQ value [10]. We also tried to use M value of 2 to get the intersection point with J-R curve. However, due to the steep slope of blunting line, it was difficult to define the intersection points for MA and AA conditions. Meanwhile, it was reported that JQ value determined using value of 2 or 4 for M is not significantly different for Category II materials with JIC in a range of 30-150 kJ/m2 [new-17]. In addition, it was reported that the slope of blunting line for determination of JQ may not affect when the ratio σYS/σUTS is close to 1 [new-18]. In case of DFW in this work, yield strength and tensile strength at 700oC is 437.0 MPa and 460.0 MPa, respectively. The calculated ratio σYS/σUTS is 0.95, which is almost value of 1. So, either of M values could be used. In this study, the measured fracture toughness value in GA and GG condition is 126.2 and 61.1 kJ/m2 (category II), respectively. Thus, in order to compare all of data set (MA, AA, GA, and GG), we chose M value of 4. The paragraph in section 3.2 was modified to explain why a value of 4 was chosen as follows.
“For fracture toughness value estimation, the blunting line was constructed by J=MσY a, in which expression a value of 2 or 4 for M was used to determine JQ value [10,17]. Meanwhile, it was reported that slope of blunting line (MσY) for determination of JQ may not affect when the ratio of yield strength (σYS) to tensile strength (σUTS) is close to 1 [18]. As summarized in Table 2, the calculation result of σYS/σUTS at 700oC is 0.95, which is almost 1. So, either of M values could be used to find JQ in this study. On the other hands, due to the steep slope of blunting line, it was difficult to define the intersection points for MA and AA conditions. Therefore, in order to compare all of data set (MA, AA, GA, and GG), we chose M value of 4 to estimate fracture toughness values.”
Thank you
Changheui Jang

Reviewer 2 Report
Review report of the manuscript
«Measurement of Fracture Toughness of Pure Tungsten Using Small-Sized Compact Tension Specimen»
Studies of the mechanical properties of pure tungsten are an urgent task in connection with its use as the main material of the first wall of fusion reactors in some international projects. Fracture toughness is one of the factors necessary for the design and prediction of structural integrity. It is known, for extremely hard metals like tungsten, the conventional test method would not work due to high stresses required to initiate crack and the very small stresses needed for driving a sharp crack. Therefore, the development of methods for determining the fracture toughness, as well as their verification, is an urgent task.
This work is a scientific and methodological study. Methodological aspects verified by ASTM standards. The geometric proportions of DCT specimen complied with general proportions of standard configuration of ASTM-E399 and ASTM-E1820. Heating of 600 ° C was used to provide the necessary ductility at the stage of preliminary crack growth. The need to protect tungsten with an inert gas from oxidation is obvious. The results of studying the morphology of fatigue preliminary cracks on samples tested in air and in argon confirmed this hypothesis.
The obtained results on the fracture toughness of pure tungsten on compact samples are verified according to previously published data.
It was established that there is no clear effect of fatigue pre-cracking, pre-cracking environment, or test environment for fracture toughness at room temperature and 400 ° C. It follows, in brittle fracture regime of DFW, the fracture toughness test using specimens with as-machined notch in air may provide reasonably accurate fracture toughness values.
On the other hand, it was found that at 700 °C, there are significant differences in measured fracture toughness values depending on fatigue pre-cracking, pre-cracking environment, and test environment
It was shown that the high oxidation rate of tungsten at 700 ° C does not allow to obtain conservative fracture toughness test results. Thus, in order to obtain reliable conservative results of the fracture toughness test at high temperatures, inert gas protection of the sample should be applied. The results obtained in this case on compact samples are most correct.
Conclusions.
The work has performed at a high scientific and technical level. Materials do not require revision for publication.
Author Response
We appreciate the valuable comments on the manuscript from the experts on measurement of fracture toughness and tungsten materials. We revised the manuscript reflecting the comments as best as possible. Here are the answers to the comments and the changes are marked in red in the revised manuscript.
Answers to REVIEWER 2’s Comments:
Ans.: We appreciate your kind evaluation on this study.
Thank you
Changheui Jang

Reviewer 3 Report
This manuscript presents the evaluation of the fracture resistance of a pure tungsten forged rod at different temperatures and environments. Some modifications are necessary.
In Figure 1(b), what do the concentrated spots represent? Besides, the contrast between red and green is poor. Is it possible to provide a better one? Regarding the tensile tests, could you mention what strain rate was used? Could you specify if the elongation was determined with the use of an extensometer? On line 84, does the z-direction mean the length of the rod? In equation 1, what is the initial crack length? And what is the meaning and dimensions of b0? You could explain the standard deviation values presented in table 3, some results do not show SD, and one value is too large. It is unclear in the manuscript why the evaluation methods at 400 ºC (E399) are different from 700 ºC (E1820). On line 146, two references are in red. Why in figure 5.b are two results for the GG condition shown? How many tensile specimens were used in each condition? And how many samples for fracture toughness determination? In figure 6, figures 6(a) and 6(b) are not in the correct place. On line 163, the fracture toughness units must be corrected.Author Response
We appreciate the valuable comments on the manuscript from the experts on measurement of fracture toughness and tungsten materials. We revised the manuscript reflecting the comments as best as possible. Here are the answers to the comments and the changes are marked in red in the revised manuscript.
Answers to REVIEWER 3’s Comments:
Ans.: The concentrated spots shown in Figure 1(b) were identified as unwashed colloidal silica used during specimen preparation. Figure 1(b) was used to provide the general microstructure of DFW used in this study. We apologize the poor quality of the photo, but we could not get the analysis time for reanalysis of EBSD due to time limitation. We indicated the nature of such spots in the revised manuscript.
Comment 3.2. Regarding the tensile tests, could you mention what strain rate was used? Could you specify if the elongation was determined with the use of an extensometer?
Ans.: The tensile tests of DFW were performed using universal testing machine (Instron 4204, load capacity 50 kN) at several temperatures with a strain rate of 2×10-4/s. The tensile elongation of samples was measured by a change of displacement without the use of an extensometer due to the small size of specimen. The following sentence was added in section 2.1 to clarify tensile test method.
“… all tests were conducted at a crosshead speed of 0.2 mm/min corresponding to a strain rate of 2 × 10-4/s. The elongation was measured by a change of displacement without the use of an extensometer due to the small size of specimen. The results of tensile tests are shown in Table 2”
Comment 3.3. On line 84, does the z-direction mean the length of the rod?
Ans.: As reviewer commented, the z-direction described in manuscript means the length of rod. In revised manuscript, the particular sentence was revised as follows to avoid confusion.
“Specimens were fabricated by electro-discharge machining (EDM) in such way that the crack plane is parallel to longitudinal direction and crack direction is in the circumferential direction of DFW rod.”
Comment 3.4. In equation 1, what is the initial crack length? And what is the meaning and dimensions of b0?
Ans.: The initial crack length (a0) is 2.7 mm. The b0 means the initial length of ligament as depicted in Figure 2. Thus, b0 is equal to 3.3 mm because width (W) of DCT is 6 mm. As following reviewer’s comment, Figure 2 was revised and specimen description was modified as follows to avoid confusion.
“The dimensions of DCT specimen are 8.1 mm of outer diameter, 6 mm of width (W), 1.5 mm of thickness (B), 2.7 mm of machined notch (a0), and 3.3 mm of remaining ligament (b0)”
Figure 5. (Figure 2 in the revised manuscript) The schematics of DCT specimen for fracture toughness test.
Comment 3.5. You could explain the standard deviation values presented in table 3, some results do not show SD, and one value is too large.
Ans.: We did not have enough specimens to test all conditions. So, some tests were done twice while some tests were done only once. The standard deviations shown in Table 3 were for 2 measured values. As reviewer pointed out, the SD value for AA condition at 700oC seems to be large. As stated in Section 3.2, for AA condition, both pre-cracking and J-R test were conducted in oxidizing environment. Therefore, the measured toughness value would be significantly affected by crack tip oxidation during pre-cracking and J-R test, which could have resulted in large scatter in JQ value compared to GA and GG conditions. However, we did not include such discussion the manuscript as test results were limited.
Comment 3.6. It is unclear in the manuscript why the evaluation methods at 400 ºC (E399) are different from 700 ºC (E1820).
Ans.: The test temperature was chosen to evaluate the different fracture behavior of DFW depending on temperature. In brittle fracture regime (below DBTT, RT and 400oC), ASTM-E399 based on linear elastic fracture mechanics (LEFM) approach was applied. Meanwhile, DFW shows elastic plastic fracture behavior at 700oC above DBTT. Thus, ASTM-E1820 was chosen to investigate fracture toughness in a view of ductile materials. We added the following sentences to Section 2.3 to clarify how the test temperatures were chosen.
“As summarized in Table 2, the DFW failed with small elongation and showed rather brittle failure at 400oC, whereas elongation was large at 600oC or higher. Therefore, it was expected that brittle fracture would be observed at RT and 400oC while ductile fracture at 700oC.”
Comment 3.7. On line 146, two references are in red.
Ans.: As reviewer commented, the color of the numbers indicating references was corrected to black.
Comment 3.8. Why in figure 5.b are two results for the GG condition shown?
Ans.: Some tests were done twice while some tests were done only once. There were two test results of GG. In the revised manuscript we indicated them as GG_01 and GG_02 in Figure R3 (Figure 5 (b) in the revised manuscript).
Comment 3.9. How many tensile specimens were used in each condition? And how many samples for fracture toughness determination?
Ans.: At least two samples were used for measuring tensile property in each test condition. For fracture toughness test, some tests were done twice while some tests were done only once. For example, two samples were used for measuring fracture toughness in each test condition except MA and AA condition at 400oC.
Comment 3.10. In figure 6, figures 6(a) and 6(b) are not in the correct place.
Ans.: We rearranged the Figure 6 to show the label of the y-axis of Figure 6 (b).
Comment 3.11. On line 163, the fracture toughness units must be corrected.
Ans.: As reviewer pointed out, the units of fracture toughness (JQ) were corrected from MPa√m to kJ/m2. Also, the related sentence in Section 3.2 was modified accordingly as follows.
“there is significant difference in measured fracture toughness between MA (JQ ~ 280 kJ/m2) and GG (JQ ~ 61 kJ/m2) conditions.”
Thank you
Changheui Jang

Round 2
Reviewer 1 Report
Thank you for the revisions, all my comments have been addressed.